# Incidence of vasa praevia: a systematic review and meta-analysis

Weiyu Zhang,[1] Tara Giacchino,[1] Pannapat Amy Chanyarungrojn  ,[1] Olivia Ionescu,[1] Ranjit Akolekar  [1,2]

¹Medway Fetal and Maternal Medicine Centre, Medway NHS Foundation Trust, Gillingham, UK
²Institute of Medical Sciences, Canterbury Christ Church University, Chatham, UK

**Correspondence to**
Professor Ranjit Akolekar;
Ranjit.Akolekar@nhs.net

## ABSTRACT

**Objectives** To derive accurate estimates of the incidence of vasa praevia (VP) in a routine population of unselected pregnancies.

**Design** Systematic review and meta-analysis.

**Data sources** A search of MEDLINE, EMBASE, CINAHL and the Cochrane database was performed to review relevant citations reporting outcomes in pregnancies with VP from January 2000 until 5 April 2023.

**Eligibility criteria for selection of studies** Prospective or retrospective cohort or population studies that provided data regarding VP cases in routine unselected pregnancies during the study period. We included studies published in the English language after the year 2000 to reflect contemporary obstetric and neonatal practice.

**Data extraction and synthesis** Two reviewers independently screened the retrieved citations and extracted data. The methodological quality of studies was assessed using the Newcastle–Ottawa Scale, and Preferred Reporting Items for Systematic reviews and Meta-Analyses was used to ensure standardised reporting of studies.

**Results** A total of 3847 citations were screened and 82 full-text manuscripts were retrieved for analysis. There were 24 studies that met the inclusion criteria, of which 12 studies reported prenatal diagnosis with a systematic protocol of screening. There were 1320 pregnancies with VP in a total population of 2 278 561 pregnancies; the weighted pooled incidence of VP was 0.79 (95% CI: 0.59 to 1.01) per 1000 pregnancies, corresponding to 1 case of VP per 1271 (95% CI: 990 to 1692) pregnancies. Nested subanalysis of studies reporting screening for VP based on a specific protocol identified 395 pregnancies with VP in a population of 732 654 pregnancies with weighted pooled incidence of 0.82 (95% CI: 0.53 to 1.18) per 1000 pregnancies (1 case of VP per 1218 (95% CI: 847 to 1901) pregnancies).

**Conclusion** The incidence of VP in unselected pregnancies is 1 in 1218 pregnancies. This is higher than is previously reported and can be used as a basis to assess whether screening for this condition should be part of routine clinical practice. Incorporation of strategies to screen for VP in routine clinical practice is likely to prevent 5% of stillbirths.

**PROSPERO registration number** CRD42020125495.

## STRENGTHS AND LIMITATIONS OF THIS STUDY

⇒ Our study provides an accurate estimate of incidence of vasa praevia based on a systematic review of studies with an a priori designed protocol registered on International Register of Systematic Reviews.

⇒ The study only includes cohort studies providing data regarding vasa praevia and control pregnancies delivered during the study period.

⇒ We only included studies that were published since the year 2000 to reflect contemporary clinical practice.

⇒ We undertook a subgroup nested analysis in studies that reported a systematic and specific policy of screening for vasa praevia in their hospital and reported both cases of vasa praevia as well as the population of control pregnancies.

⇒ A limitation of our study is that we did not provide separate incidence for singleton and multiple pregnancies.

traversing the amniotic membranes in the lower uterine segment in close proximity to the internal cervical os.[1–5] These unsupported fetal vessels can be damaged either during the antenatal period or in labour leading to severe hypovolaemic shock and haemorrhagic fetal death.[4 6 7] There is considerable evidence suggesting that prenatal diagnosis of VP improves perinatal survival and lack of antenatal detection is associated with a high risk of stillbirths, neonatal deaths and an increased risk of hypoxic morbidity in the survivors.[8–13] Despite the compelling evidence in favour of prenatal diagnosis in improving pregnancy outcomes in pregnancies with VP, there are currently no recommendations for screening for VP in routine clinical practice.[14] The UK National Screening Committee examined the evidence with regard to screening for VP and reported that there was low volume of evidence on the epidemiology of VP or its outcomes in the UK to justify screening.[15]

A systematic review examined the incidence of VP and reported that there was a huge variation in studies reporting VP

## INTRODUCTION

Vasa praevia (VP) is defined as presence of fetal blood vessels, arterial or venous, unsupported by placental tissue or umbilical cord

incidence ranging from 0.19 to 2.21 per 1000 pregnancies (1 per 452–1 per 5882 pregnancies) with a mean estimated incidence of 0.6 per 1000 pregnancies (1 per 1666 pregnancies).[16] This systematic review had significant heterogeneity and variation as it included not just cohort studies but case–control studies and those from high-risk populations. The estimated incidence in this systematic review is likely to have been underestimated as many of the included studies reported data from 80s and 90s when VP was more likely to be an incidental rather than an intentional diagnosis. In a recent large, prospective, observational screening study of more than 26 000 pregnancies, we demonstrated the impact of routine screening for VP based on a two-stage screening strategy and reported that the incidence of VP was about 1 in 1277 pregnancies.[12] The protocol states that stage 1 includes checking for umbilical cord insertion and placental location, those that are velamentous cord insertions in lower segment and those with a low-lying placenta are classed as at risk; and stage 2 would include a transvaginal assessment with colour Doppler ultrasound to check for presence of any fetal vessels in the vicinity of the cervix for a diagnosis of VP.[12]

The objective of this study was to undertake a systematic review of literature and undertake a meta-analysis to estimate an accurate incidence of VP in a routine population of unselected pregnancies based on analysis of studies that reported a policy of routinely screening for VP.

## METHODS
### Data sources and search strategy
This systematic review and meta-analysis was undertaken based on an a priori designed study protocol recommended for systematic reviews and meta-analyses.[17] The study protocol for the systematic review was registered in advance with the PROSPERO International Prospective Register of Systematic Reviews (registration number: CRD42020125495) as a study to investigate perinatal outcomes as well as incidence of VP. An electronic search of MEDLINE, EMBASE and the Cochrane Library was carried out on 30 October 2022 and repeated on 5 April 2023 using combinations of the relevant Medical Subject Heading terms, keywords and word variants for "vasa praevia", "abnormal cord insertion", "velamentous cord", marginal cord", "bilobed placenta" and "succenturiate lobe". The search and selection criteria were restricted to studies reported in the English language and are outlined in online supplemental table S1.

### Selection criteria for studies and data extraction
The studies eligible for inclusion in this study were prospective or retrospective cohort studies and population-based studies reporting cases with VP in which the authors reported data about the total screened population during the study period. We only included studies published after 2000 to reflect contemporary clinical practice. We did a nested subanalysis for studies which reported in

their methodology that they had a routine screening policy for assessing umbilical cord insertion and VP. The citations were examined by two reviewers (WZ and PAC) to produce a list of relevant studies after exclusion of duplicates; those that did not fit selection criteria after review of title and abstract; those that were case reports, letter to editors, review articles or conference abstracts; and those that were case series or case–control studies. The reference lists of relevant articles were searched, and any inconsistencies were resolved by discussion with an internal third reviewer (RA).

The citations retrieved following this search strategy were examined for relevance to this study to investigate incidence based on above eligibility criteria and in each study, we extracted information about authors, study location, years of enrolment for cases and controls, study design, sample size, whether the study was single or multicentre, whether the study was in high-risk population or unselected general pregnant population, and whether there was a routine policy of screening for umbilical cord insertion and VP during the study period in the reported studies. Data extracted for each study were inputted into contingency tables.

### Quality assessment
The methodological quality of studies included in the review was assessed using the Newcastle–Ottawa Scale (NOS).[18] We used PRISMA (Preferred Reporting Items for Systematic reviews and Meta-Analyses) to ensure standardised reporting of studies in our systematic review. The PRISMA statement for this study included a checklist and a flow chart to allow uniform and transparent reporting of the systematic review and meta-analysis.[19]

### Estimation of summary statistics for incidence of VP
Data were entered in contingency tables for cases and control pregnancies to calculate weighted pooled incidence (95% CI). The incidence of VP was reported as total number of cases per 1000 pregnancies as well as total number of pregnancies for every 1 case of VP. Summary statistics (95% CI) from each study were then combined to obtain a pooled estimate which was calculated as a weighted average of the individual study estimates. The pooled summary statistics were estimated using random-effects model (REM) to allow for assessment of between-study variability in results by weighting studies using a combination of their own variance as well as the between-study variance.[20] The heterogeneity between studies was estimated using Cochran's heterogeneity statistic Q and the $I^2$ statistic. Regression analysis was undertaken to examine factors providing a significant contribution to the prediction of incidence of VP. We also estimated the potential impact of introducing a policy for screening for VP in preventing stillbirths in the UK based on annual number of total births and stillbirths reported by the Office of National Statistics, National Records of Scotland and Northern Ireland Statistics and Research Agency.[21–23] The statistical software package StatsDirect V.2.7.9

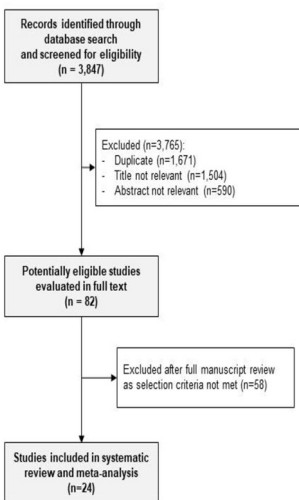

**Figure 1** Flow chart demonstrating process of selection of studies included in the systematic review.

(StatsDirect, Cheshire, UK) and MedCalc Statistical Software V.16.4.3 (MedCalc Software, Ostend, Belgium) were used for data analysis.

### Patient and public involvement
No patients were involved in setting the research question, and in developing plans for design, interpretation, reporting or implementation of the study.

## RESULTS
### Data search results
The electronic search from the databases yielded 3847 potential citations; of these, we excluded 1671 citations as they were duplicates, 1504 after review of study title and 590 citations after review of the abstract. We retrieved 82 manuscripts in full text for detailed assessment and excluded a further 58 studies that did not meet the selection criteria, thus finally including 24 studies for systematic review and meta-analysis. The search criteria are outlined in online supplemental table S1, and the study selection process is outlined in the flow chart in figure 1.

### Characteristics of studies included in the systematic review
There were 24 studies included in the systematic review,[8 10–12 24–43] which provided data regarding incidence of VP. There were 7 cohort studies published in the first decade (2000–2010),[8 24–29] and 17 studies in the second decade (2011–2020).[10–12 30–43] The published studies were from across different geographical areas: eight studies from the USA,[8 10 24 31 32 34 36 38] five studies from Japan,[27 29 30 33 43] six from Europe,[12 26 35 37 40 41] two each from Australia[11 39] and Israel,[25 28] and one from China.[42] All 24 studies reported the total screened population during the study period and the number of cases of VP detected, thus providing a basis of estimating incidence of VP. Screening studies which report a policy-based assessment of an unselected cohort of pregnancies are more likely to reflect the true incidence of VP, compared

with studies that included an incidental diagnosis of VP. Therefore, we identified studies that described a specific protocol for assessment of all pregnancies during the study period for prenatal diagnosis of VP after excluding those that had a variable protocol for identification of at-risk pregnancies; those that were carried out in a specific high-risk population such as those with velamentous cord insertion,[37] succenturiate lobe[27] and in vitro fertilisation pregnancies[25]; or those who were only delivered by caesarean section.[36] After exclusion of above studies, we identified 12 studies,[10–12 26 31 34 35 38–41 43] in which the authors reported that they had a systematic protocol to document the placental location and umbilical cord insertion at the first or second trimester scan; those that were suspected to have VP were further scanned with transvaginal ultrasound to confirm the diagnosis. In the 12 screening protocol-based studies included, the authors reported confirmation of the presence of VP postnatally (online supplemental table S2).

### Assessment of quality and heterogeneity of studies
The methodological quality of studies included in this systematic review was assessed using the NOS. The rating of the included studies according to the NOS based on selection, comparability and outcome is shown in online supplemental table S3. The PRISMA guidance was followed for reporting this meta-analysis and is reported in online supplemental table S4. The funnel plot of included studies demonstrating the publication bias is shown in online supplemental figure F1.

### Incidence of VP
There were 24 studies that reported data about cases with VP with a corresponding cohort of total number of unselected pregnancies screened in a routine population during the study period.[8 10–12 24–43] There were 1320 pregnancies with VP in a total population of 2278561 pregnancies. The weighted pooled incidence of VP was 0.79 (95% CI: 0.59 to 1.01) per 1000 pregnancies screened, which corresponded to an incidence of 1 case of VP per 1271 (95% CI: 990 to 1692) pregnancies (table 1).

There were 12 studies that reported data regarding incidence of VP along with the total pregnancies screened based on a specific screening protocol.[10–12 26 31 34 35 38–41 43] There were 395 pregnancies with VP in a total population of 732654 pregnancies. The weighted pooled incidence of VP was 0.82 (95% CI: 0.53 to 1.18) per 1000 pregnancies screened, which corresponded to an incidence of 1 case of VP per 1218 (95% CI: 847 to 1901) pregnancies (table 2 and figure 2).

Regression analysis demonstrated that in prediction of the incidence of VP, there was a significant independent contribution from the publication date of the study, whether the study was published in the first decade (2000–2010) or in the second decade (2011–2022) (p<0.011), but there was no significant association with the country in which the study was conducted (p=0.408) (R²=0.208; p<0.05). There was a significant increase in incidence of

**Table 1** Results from meta-analysis using a random-effects model to estimate summary statistics for incidence (95% CI) of vasa praevia (VP) based on studies that reported VP cases along with data regarding an unselected cohort of pregnancies during the study period

| Author (year) | n/N | n/1000 pregnancies | 1/n pregnancies |
|---|---|---|---|
| Lee et al (2000)[24] | 18/93 874 | 0.19 (0.11 to 0.30) | 5208 (3300 to 8772) |
| Catanzarite et al (2001)[8] | 11/33 208 | 0.33 (0.17 to 0.59) | 3201 (1686 to 6061) |
| Schachter et al (2002)[25] | 12/72 818 | 0.17 (0.09 to 0.29) | 6061 (3472 to 11 737) |
| Baulies et al (2007)[26] | 9/12 063 | 0.75 (0.34 to 1.42) | 1340 (706 to 2933) |
| Suzuki and Igarashi (2008)[27] | 3/7713 | 0.39 (0.08 to 1.14) | 2571 (877 to 12 469) |
| Smorgick et al (2010)[28] | 19/110 684 | 0.17 (0.10 to 0.27) | 5814 (3731 to 9709) |
| Hasegawa et al (2010)[29] | 10/4532 | 2.21 (1.06 to 4.05) | 452 (247 to 943) |
| Kanda et al (2011)[30] | 10/5131 | 1.95 (0.94 to 3.58) | 513 (279 to 1070) |
| Rebarber et al (2013)[31] | 24/27 573 | 0.87 (0.56 to 1.29) | 1149 (775 to 1792) |
| Bronsteen et al (2013)[32] | 74/182 554 | 0.41 (0.32 to 0.51) | 2469 (1965 to 3145) |
| Hasegawa et al (2015)[33] | 21/8176 | 2.57 (1.59 to 3.92) | 389 (255 to 629) |
| Catanzarite et al (2016)[10] | 96/100 481 | 0.96 (0.77 to 1.17) | 1047 (857 to 1292) |
| Kulkarni et al (2017)[34] | 35/56 000 | 0.63 (0.44 to 0.87) | 1600 (1151 to 2299) |
| Nohuz et al (2017)[35] | 8/18 152 | 0.44 (0.19 to 0.87) | 2268 (1152 to 5263) |
| Sullivan et al (2017)[11] | 63/294 045 | 0.21 (0.17 to 0.27) | 4673 (3650 to 6061) |
| Yeaton-Massey et al (2019)[36] | 586/945 950 | 0.62 (0.57 to 0.67) | 1616 (1488 to 1754) |
| Derisbourg et al (2019)[37] | 4/1620 | 2.47 (0.67 to 6.31) | 405 (158 to 1486) |
| Klahr et al (2019)[38] | 61/37 236 | 1.64 (1.25 to 2.10) | 610 (476 to 800) |
| La et al (2020)[39] | 19/56 045 | 0.34 (0.20 to 0.53) | 2950 (1890 to 4902) |
| Zhang et al (2020)[12] | 21/26 830 | 0.78 (0.49 to 1.20) | 1277 (836 to 2062) |
| Gross et al (2021)[40] | 21/5905 | 3.56 (2.20 to 5.43) | 281 (184 to 455) |
| Sutera et al (2021)[41] | 24/89 600 | 0.27 (0.17 to 0.40) | 3731 (2506 to 5814) |
| Liu et al (2021)[42] | 157/79 647 | 1.97 (1.68 to 2.30) | 508 (435 to 595) |
| Kamijo et al (2022)[43] | 14/8723 | 1.60 (0.88 to 2.69) | 625 (372 to 1139) |
| **Pooled analysis** | **1320/2 278 561** | **0.79 (0.59 to 1.01**) | **1271 (990 to 1692**) |
| Cochran's Q (df) | 550.11 (23) | | |
| $I^2$ (inconsistency) (%) (95% CI) | 95.8 (94.7 to 96.7) | | |
| Egger bias (p value) | 2.1799 (0.1692) | | |

VP from studies published in the second decade (2011–2022) compared with first decade (2000–2010) (OR: 2.60; 95% CI: 2.08 to 3.25) (p<0.01).

### Potential impact on prevention of stillbirths in the UK

There were a total of 697 551 births in the UK with 2866 (0.41%) stillbirths in 2021. Given an incidence of VP of 1 in 1218 pregnancies, there would be an estimated 573 pregnancies with VP per year in the UK. Based on estimates from the systematic review and meta-analysis reporting perinatal survival in pregnancies with VP, the live birth rate in pregnancies with VP with and without prenatal diagnosis is 98.6% (95% CI: 96.7% to 99.7%) and 72.1% (95% CI: 50.6% to 89.4%), respectively; therefore, the expected number of stillbirths in pregnancies with VP in those with and without prenatal diagnosis would be 8 (95% CI: 2 to 19) and 160 (95% CI: 61 to 283). Therefore, the potential impact of introducing a policy

of screening for VP in clinical practice could contribute to prevention of 5.30% (95% CI: 2.11% to 9.89%) of stillbirths in the UK. This is based on the assumption that VP is causal to the fetal death but in some pregnancies with VP, other risk factors and/or pregnancy complications may also play a role.

### DISCUSSION
### Principal findings of the study

The findings of our systematic review and meta-analysis demonstrate that the incidence of VP in an unselected population of pregnancies is 1 in 1218 pregnancies. The estimated incidence of VP in our study is based on weighted pooled estimates calculated first, from meta-analysis of studies published in the last 20 years to reflect contemporary obstetric and neonatal practice

**Table 2** Results from meta-analysis using a random-effects model to estimate summary statistics for incidence (95% CI) of vasa praevia (VP) based on screening cohort studies that reported a specific policy of screening for VP during the study period

| Author (year) | n/N | n/1000 pregnancies | 1/n pregnancies |
|---|---|---|---|
| Baulies et al (2007)[26] | 9/12 063 | 0.75 (0.34 to 1.42) | 1340 (706 to 2933) |
| Rebarber et al (2013)[31] | 24/27 573 | 0.87 (0.56 to 1.29) | 1149 (775 to 1792) |
| Catanzarite et al (2016)[10] | 96/100 481 | 0.96 (0.77 to 1.17) | 1047 (857 to 1292) |
| Kulkarni et al (2017)[34] | 35/56 000 | 0.63 (0.44 to 0.87) | 1600 (1151 to 2299) |
| Nohuz et al (2017)[35] | 8/18 152 | 0.44 (0.19 to 0.87) | 2268 (1152 to 5263) |
| Sullivan et al (2017)[11] | 63/294 045 | 0.21 (0.17 to 0.27) | 4673 (3650 to 6061) |
| Klahr et al (2019)[38] | 61/37 236 | 1.64 (1.25 to 2.10) | 610 (476 to 800) |
| La et al (2020)[39] | 19/56 045 | 0.34 (0.20 to 0.53) | 2950 (1890 to 4902) |
| Zhang et al (2020)[12] | 21/26 830 | 0.78 (0.49 to 1.20) | 1277 (836 to 2062) |
| Gross et al (2021)[40] | 21/5905 | 3.56 (2.20 to 5.43) | 281 (184 to 455) |
| Sutera et al (2021)[41] | 24/89 600 | 0.27 (0.17 to 0.40) | 3731 (2506 to 5814) |
| Kamijo et al (2022)[43] | 14/8723 | 1.60 (0.88 to 2.69) | 625 (372 to 1139) |
| **Pooled analysis** | **395/732 654** | **0.82 (0.53 to 1.18)** | **1218 (847 to 1901)** |
| Cochran's Q (df) | 222.18 (11) | | |
| $I^2$ (inconsistency) (%) (95% CI) | 95.1 (92.9 to 96.5) | | |
| Egger bias (p value) | 6.1508 (0.0092) | | |

and second, by inclusion of those studies that reported a specific policy of assessment of placental location and umbilical cord insertion for prenatal diagnosis of VP in their routine practice. The incidence of VP was not affected by the study location or where the study was carried out, reflecting that in centres that have a specific protocol for assessment of VP as described above, there is no significant variation in reported incidence. Our study demonstrates that if screening for prenatal diagnosis of VP was introduced in routine clinical practice, it can potentially prevent 5% of stillbirths.

## Comparison with previous studies

The results of our study demonstrate that VP is not an uncommon obstetric complication and has an incidence

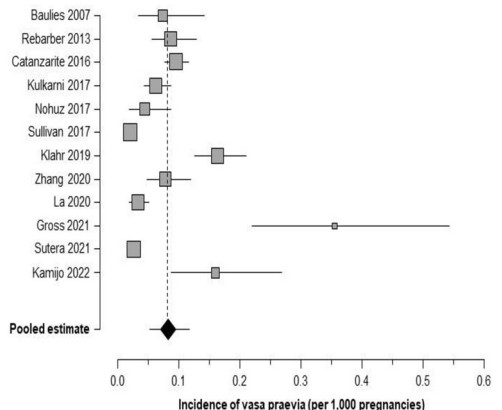

**Figure 2** Forest plot demonstrating incidence of vasa praevia in studies reporting results of routine screening in low-risk unselected pregnancies. Incidence (%; 95% CIs) is reported as number of cases per 1000 pregnancies.

of 1 in 1218 pregnancies, which is higher than that reported in previous studies.[16] A previous systematic review reported a wide variation in the incidence of VP from 1 per 452–5882 pregnancies with a mean incidence of 1 in 1666 pregnancies. The authors included a wide range of studies including case–control studies, retrospective cohort studies, studies carried out on high-risk populations (such as those with pregnancies conceived with in vitro fertilisation, studies examining outcomes in pregnancies with bilobed placenta or in those that had a velamentous cord insertion), those that reported an incidental diagnosis of VP rather than cases detected in an unselected population based on a systematic protocol of screening and inclusion of studies that did not report whether there was postnatal confirmation of prenatal findings of VP, thus leading to a wide heterogeneity in the results and potentially underestimation of the incidence. Moreover, the authors included studies reporting data from 80s and 90s, which are more likely to be incidentally diagnosed cases of VP rather than as a result of intentional screening.

## Implications for clinical practice

Our study highlights that the incidence of VP is much higher than is previously reported. The results of our study provide an accurate estimate of the incidence which can, for the basis for further studies, investigate whether VP screening should be incorporated in routine practice as not only it is more common than previously believed, but more importantly, prenatal diagnosis can prevent adverse outcomes associated with VP. There is robust and unequivocal evidence from several studies, including our cohort study, suggesting that prenatal diagnosis of VP

is associated with high rates of live births and perinatal survival.[10–12 30 34 35 40–42] A recent systematic review and meta-analysis by our group summarised and confirmed the results from previous studies demonstrating that prenatal diagnosis of VP is associated with a 99% rate of perinatal survival, but a lack of an antenatal diagnosis increases the risk of a perinatal hypoxic morbidity 50-fold and that of perinatal death by 25-fold.[13] There are few other examples in obstetric practice which demonstrate such a significant and stark impact of prenatal diagnosis on improving pregnancy outcomes such as the prenatal diagnosis of VP. There are 12 relatively recent studies,[10–12 26 31 34 35 38–41 43] which report that systematic and routine screening for VP is feasible in routine clinical practice; in fact, our recent prospective observational study of more than 26 000 pregnancies demonstrates that a two-stage screening strategy for VP can be easily implemented in routine clinical practice and such a policy is associated with a high rate of prenatal diagnosis and good perinatal outcomes.[12] The recommendations for screening for VP in routine practice are hindered by evidence about accurate incidence in routine pregnant population. Our study should pave the way for further studies that investigate strategies of introduction screening for VP in routine pregnancies.

## Strengths and limitations

The strength of our study is that it summarises results of all studies reported in the last two decades and provides accurate summary estimates for incidence of VP in current obstetric practice. We used a standardised methodology for conducting this systematic review and meta-analysis with design of an a priori protocol, registration with the international register at PROSPERO, a comprehensive search strategy, appropriate quality assessment of studies included in systematic review with NOS and standardised reporting of systematic review with PRISMA checklist. The limitations of our study relate to standard biases associated with meta-analysis such as inclusion of studies with different sample sizes, methodology and design which may introduce heterogeneity in the analysis. However, as a means to minimise the effects of such variation and heterogeneity, we reviewed all these studies in detail to select and identify those studies that provide data from low-risk unselected pregnancies rather than high-risk populations. This allowed us to do a nested meta-analysis to estimate an accurate weighted pooled summary statistic, which is more reflective of true population incidence rather than just a pooled estimate from all reported studies. We used an REM to mitigate the impact of heterogeneity as this model allows for assessment of between-study variability in results by weighing studies using a combination of their own variance as well as the between-study variance. Furthermore, we performed regression analyses to examine temporal and spatial trends in VP incidence such as the publication dates as well as the location of where the studies were carried out. A confounding factor and a limitation of our study is that we did not exclude multiple pregnancies, and this incidence is the total incidence in singleton and multiple pregnancies; the incidence in multiple pregnancies could potentially be higher but requires further studies to investigate this.

## Conclusions

VP is not an uncommon obstetric complication. The incidence of VP in an unselected population is 1 in 1218 pregnancies, which is considerably higher than previously reported estimates. Further research studies should be undertaken to model and investigate strategies for incorporating prenatal screening for VP in routine clinical practice, given the feasibility of screening and major impact on prevention of stillbirths and hypoxic morbidity with prenatal diagnosis of VP.

**Acknowledgements** We are grateful to the Medway Library and Knowledge Service for their help and assistance with search strategy. This study is a part of the PhD thesis of Weiyu Zhang for the Canterbury Christ Church University.

**Contributors** WZ, PAC, TG and OI contributed to literature search, study selection, data extraction and assessment of study quality. WZ and RA wrote the initial and final drafts of the manuscript. RA conceived the study and supervised the data extraction, discussed studies for inclusion following initial review and agreed with the final draft of the manuscript for submission. All authors reviewed the initial and final drafts of the manuscript and contributed to the intellectual content. RA acted as guarantor responsible for the conduct of the study, overall content and decision to publish.

**Funding** The authors have not declared a specific grant for this research from any funding agency in the public, commercial or not-for-profit sectors.

**Competing interests** None declared.

**Patient and public involvement** Patients and/or the public were not involved in the design, or conduct, or reporting, or dissemination plans of this research.

**Patient consent for publication** Not required.

**Ethics approval** No ethical approval was required for this study.

**Provenance and peer review** Not commissioned; externally peer reviewed.

**Data availability statement** All data relevant to the study are included in the article or uploaded as supplemental information.

**ORCID iDs**
Pannapat Amy Chanyarungrojn http://orcid.org/0000-0001-5139-8678
Ranjit Akolekar http://orcid.org/0000-0001-7265-5442

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
