## [Reviewer comments · BMJ Open]

ARTICLE DETAILS

TITLE (PROVISIONAL)	Incidence of Vasa praevia: A systematic review and meta-analysis
AUTHORS	Zhang, Weiyu; Giacchino, Tara; Chanyarungrojn, Pannapat Amy; Ionescu, Olivia; Akolekar, Ranjit

VERSION 1 – REVIEW

REVIEWER	A Herrera, Christina Orlando Health Winnie Palmer Hospital for Women & Babies
REVIEW RETURNED	31-May-2023

GENERAL COMMENTS	This SR/MA manuscript is a nice presentation of the contemporary incidence of vasa previa. The study design and methodology are thoroughly presented. The results are clearly explained and justified. The clinical implications are also nicely described. The potential impact on stillbirth (with proper screening) is noteworthy and underscored. As the authors nicely point out, a contemporary update to the incidence is necessary to emphasize the importance of screening. A few specific considerations are listed below. 1. Consider specifying in the abstract conclusion that this number is referring to stillbirths in the UK.2. There is a reference (#12) to a 2-stage screening protocol previously published. Consider elaborating on that protocol (1-2 sentences) in this paper as it is relevant information.3. I cannot comment on the statistical analyses for this SR/MA and would recommend expert review of that portion.4. Consider attempting to further highlight the clinical impact of this study in the abstract and body of the paper. At first glance, a reader may think this is just an incidence report without much clinical meaning. However, the higher incidence than previously thought is quite relevant and important.
--

REVIEWER	Pecks, Ulrich Universitätsklinikum Schleswig-Holstein Campus Kiel
REVIEW RETURNED	04-Jun-2023

GENERAL COMMENTS	This is a systematic review and meta-analysis of the incidence of vasa praevia (VP). The authors aimed to estimate the incidence of vasa praevia in a general obstetric population and from this to calculate the potential reduction in stillbirths through screening. The meta-analysis included population-based studies as well as prospective and retrospective cohorts in which screening for VP was reported. The authors reviewed a total of 3847 citations, 24 of which met the inclusion criteria. Studies included a total of 2 million women with an overall incidence of 0.079%. Subanalyses of 12 studies reporting exact screening measures in the first or
---

	second trimester yielded similar results. The authors concluded that the introduction of screening measures for VP would lead to a 5.8% decrease in stillbirths in the United Kingdom. The study was thoroughly conducted. Scientific quality is high. The abstract correctly reflects the content of the study. The methodology is correctly described. The study was registered with PROSPERO primarily for a recent meta-analysis on perinatal outcome after VP. However, there are some aspects that could improve the manuscript and should be carefully reviewed. A) Authors should indicate whether they exclude or include multiple pregnancies, and if so, this should be included as a potential confounder. B) In addition, authors should consider previous cesarean section (CS) as a confounding factor because CS is a risk factor for VP and the frequency of CS may vary over time or across studies. C) Authors should provide more details on the sensitivity and specificity for detecting VP and whether diagnoses were confirmed after delivery in the underlying studies. D) Page 6 line 128. Perhaps it would be useful to describe PICOS in detail at least for the 12 studies in which screening measures were introduced. E) Page 9 line 230. Authors stated that screening for VP could contribute to prevention of 5.58% of stillbirth in the UK. Strictly speaking, it must be 5.3%, because the authors had to subtract the women in whom CP was missed by screening. F) The authors should discuss that the calculated 5.3% reduction is valid only under the assumption that VP is indeed causal and not merely incidental to the stillbirths counted. G) Page 10 line 256. The authors should consider that the lower VP rate in Ref 16 may be due to the lower CS rate in the 1980s or 1990s and that the incidence of VP may have actually increased over time. This should be discussed in line with Ref 16.. H) Please note that numbers of tables and figures partly are incorrect. Suppl. Table 2 is named ST1. Suppl. Table 3 is named ST2
--	---

REVIEWER	Milic, Jelena Erasmus MC
REVIEW RETURNED	23-Jun-2023

GENERAL COMMENTS	Dear authors, with great pleasure I read your study. Congratulations on a well-structured methodology of systematic reviewing. Also, the incidence of VP is not sufficiently studied so far. This gives an added value to your research. I have few minor revision suggestions that I would like you to consider I enclose them bellow. Looking forward to seeing a revised version ABSTRACT Page 3, Line 37 Data extraction and synthesis: Two reviewers independently screened the retrieved 37 citations and extracted data... What have you done in case of disagreement between 2 independent assessments? Was there a 3rd appointed non previously involved expert in the topic who would made an arbitrary decision in such case. To clarify, an arbitrary decision explains decisions made or actions taken that are not necessarily based on established facts, but instead based in large part on expert opinions. Arbitrary decisions do not reflect accepted exclusively practice precedence, nor are they made with regards to existing facts or established circumstances. Why is this important to mention
--

There are several approaches to study inclusion: using a single reviewer, dual review with internal resolution, dual review with third party resolution, or crowdsourcing with three or more reviewers. . If you have used just dual review with internal resolution but there was no need for 3rd party resolution, make a mention. In case you did not mention in limitations:  • Introducing some bias, • Requires highly trained expert in topic to start the search • Time-consuming while agreeing on resolution STRENGTHS AND LIMITATIONS OF THE STUDY Page 4, Lines 110-113 Even though it is most probably clear to the authors what is considered to be the strength and what the opposite, to the reader some of the bullet points remain ambiguous, Either split in strength and limitations bullet points, or state the strength or limitation side in each statement as you do later on at Page 10, line 287 Data sources and search strategy Page 6, Lines 110-113 You did not define or predefine the MeSH terms you used to search by PICO criterion or you do not mention this, In both cases please make a statement about this and add it here. For more info on PICO look here https://mcw.libguides.com/EBM/PICO#:~:text=PICO%20stands%20for%20patient%2Fpopulation%2C%20intervention%2C%20comparison%20and%20outcomes.&text=Who%20is%20your%20patient%3F&text=What%20do%20you%20plan%20on%20doing%20for%20the%20patient%3F&text=What%20alternative%20are%20you%20considering%3F Page 10, Lines 272-273 ...prospective cohort screening study Please rephrase to our (our own is a pleonasm) cohort study comparing the outcome of the screening of incidence of... Stated as it is, introduces confusion about the study design What is a screening study? More precisely, a screening trial is designed to assess or test the efficacy of devices, techniques, procedures, or tests to detect the presence of disease or condition (or risk factors for a disease or condition), usually before there are any symptoms. What is a cohort study? (KOH-hort STUH-dee) A research study that compares a particular outcome (such as lung cancer) in groups of individuals who are alike in many ways but differ by a certain characteristic (for example, female nurses who smoke compared with those who do not smoke). As your study probably compares the outcome of the screening – please rephrase Page 10, Line 286 Add a paragraph about future research directions

VERSION 1 – AUTHOR RESPONSE

Reviewer Reports:

Reviewer: 1

Dr. Christina A Herrera, Orlando Health Winnie Palmer Hospital for Women & Babies

Comments to the Author:

This SR/MA manuscript is a nice presentation of the contemporary incidence of vasa previa. The study design and methodology are thoroughly presented. The results are clearly explained and justified. The clinical implications are also nicely described. The potential impact on stillbirth (with proper screening) is noteworthy and underscored. As the authors nicely point out, a contemporary update to the incidence is necessary to emphasize the importance of screening. A few specific considerations are listed below.

Response: Thank you.

1. Consider specifying in the abstract conclusion that this number is referring to stillbirths in the UK.

Response: Thank you. The rate of stillbirths in UK and USA is not dissimilar. Although we have presented the analysis in the results section based on births in the UK but in a general sense, it reflects impact on stillbirth prevention in the developed world and therefore would also apply to the USA. We have estimated the potential percentage reduction in stillbirths given the incidence of VP, the rate of livebirths and survival based on systematic reviews and the rate of stillbirths along with confidence intervals, so this should be generally applicable to the developed world with similar healthcare settings and does not reflect stillbirths in a resource poor setting. We are happy to state this in the abstracts but will leave it to editorial discretion.

2. There is a reference (#12) to a 2-stage screening protocol previously published. Consider elaborating on that protocol (1-2 sentences) in this paper as it is relevant information.

Response: Thank you. We have now explained this briefly in the introduction in 1-2 sentences.

3. I cannot comment on the statistical analyses for this SR/MA and would recommend expert review of that portion.

Response: Thank you for commenting on other aspects of the paper. Noted.

4. Consider attempting to further highlight the clinical impact of this study in the abstract and body of the paper. At first glance, a reader may think this is just an incidence report without much clinical meaning. However, the higher incidence than previously thought is quite relevant and important.

Response: Thank you indeed. We agree entirely. We have further highlighted this in the abstract and the body of the paper.

Reviewer: 2

Dr. Ulrich Pecks, Universitätsklinikum Schleswig-Holstein Campus Kiel

Comments to the Author:

This is a systematic review and meta-analysis of the incidence of vasa praevia (VP). The authors aimed to estimate the incidence of vasa praevia in a general obstetric population and from this to calculate the potential reduction in stillbirths through screening. The meta-analysis included population-based studies as well as prospective and retrospective cohorts in which screening for VP was reported. The authors reviewed a total of 3847 citations, 24 of which met the inclusion criteria. Studies included a total of 2 million women with an overall incidence of 0.079%. Subanalyses of 12 studies reporting exact screening measures in the first or second trimester yielded similar results. The authors concluded that the introduction of screening measures for VP would lead to a 5.8% decrease in stillbirths in the United Kingdom.

The study was thoroughly conducted. Scientific quality is high. The abstract correctly reflects the content of the study. The methodology is correctly described. The study was registered with PROSPERO primarily for a recent meta-analysis on perinatal outcome after VP. However, there are some aspects that could improve the manuscript and should be carefully reviewed.

A) Authors should indicate whether they exclude or include multiple pregnancies, and if so, this should be included as a potential confounder.

Response: Thank you. We did not exclude multiple pregnancies. We have now added a statement in study limitations about this.

B) In addition, authors should consider previous cesarean section (CS) as a confounding factor because CS is a risk factor for VP and the frequency of CS may vary over time or across studies.

Response: Thank you. We have not come across many studies which state that previous caesarean section is a major risk factor. In fact, a systematic review examining risk factors associated with VP did not state previous CS as a risk factor nor was this identified in our own cohort study.

C) Authors should provide more details on the sensitivity and specificity for detecting VP and whether diagnoses were confirmed after delivery in the underlying studies.

Response: Thank you. Our study is about incidence of VP. We have therefore only focussed on cases of VP diagnosed antenatally and confirmed postnatally. We excluded VP cases that were suspected and subsequently resolved. We have reviewed all the 12 screening studies that had a systematic screening protocol and all studies stated in their methods that postnatal confirmation of VP was made. We have now added a statement in the characteristics of studies in the results section to reflect that.

D) Page 6 line 128. Perhaps it would be useful to describe PICOS in detail at least for the 12 studies in which screening measures were introduced.

Response: Thank you. Characteristics of the included studies are already described in considerable detail in the second paragraph of the results, but we have also now added a Supplementary Table S4 to describe these as well.

E) Page 9 line 230. Authors stated that screening for VP could contribute to prevention of 5.58% of stillbirth in the UK. Strictly speaking, it must be 5.3%, because the authors had to subtract the women in whom CP was missed by screening.

Response: Thank you. We believe the statistics we have stated in results are correct. Based on our incidence of VP, there would be 573 pregnancies with VP annually in UK. Assuming a 98.6% livebirth rate in pregnancies with prenatal diagnosis, there would be 565 live births and 8 stillbirths. Assuming a livebirth rate of 72.1% in those without prenatal diagnosis, there would be 413 live births and 160 stillbirths. Given that there are annually reported 2,866 stillbirths in the UK, those due to VP would be 5.58% (160/2866).

F) The authors should discuss that the calculated 5.3% reduction is valid only under the assumption that VP is indeed causal and not merely incidental to the stillbirths counted.

Response: Thank you. We agree that there may be other causes of stillbirths in pregnancies with VP and it is true that the calculated 5.58% is only valid if VP is causal. We have now added a statement to this effect in the results.

G) Page 10 line 256. The authors should consider that the lower VP rate in Ref 16 may be due to the lower CS rate in the 1980s or 1990s and that the incidence of VP may have actually increased over time. This should be discussed in line with Ref 16..

Response: Thank you. We believe that the reason for the lower incidence of VP in reference 16 is not necessarily because of the rising rate of CS since the 80s but other causes, which we have outlined in our paragraph in comparison with other studies in which we discuss this in detail. "The authors included a wide range of studies including case-control studies, retrospective cohort studies, studies carried out on high-risk populations (such as those with pregnancies conceived with in vitro fertilisation, studies examining outcomes in pregnancies with bilobed placenta or in those that had a velamentous cord insertion), those that reported an incidental diagnosis of VP rather than cases detected in an unselected population based on a systematic protocol of screening and inclusion of studies that did not report whether there was postnatal confirmation of prenatal findings of VP, thus leading to a wide heterogeneity in the results and potentially underestimation of the incidence."

H) Please note that numbers of tables and figures partly are incorrect. Suppl. Table 2 is named ST1. Suppl. Table 3 is named ST2

Response: Thank you. We have now amended this.

Reviewer: 3

Dr. Jelena Milic, Erasmus MC, Institut za javno zdravlje Srbije Dr Milan Jovanovic Batut

Comments to the Author:

Dear authors,

with great pleasure I read your study. Congratulations on a well-structured methodology of systematic reviewing. Also, the incidence of VP is not sufficiently studied so far. This gives an added value to your research. I have few minor revision suggestions that I would like you to consider I enclose them below.

Looking forward to seeing a revised version

Response: Thank you.

ABSTRACT

Page 3, Line 37

Data extraction and synthesis:

Two reviewers independently screened the retrieved 37 citations and extracted data...

What have you done in case of disagreement between 2 independent assessments? Was there a 3rd appointed non previously involved expert in the topic who would made an arbitrary decision in such case. To clarify, an arbitrary decision explains decisions made or actions taken that are not necessarily based on established facts, but instead based in large part on expert opinions. Arbitrary decisions do not reflect accepted exclusively practice precedence, nor are they made with regards to existing facts or established circumstances. Why is this important to mention? There are several approaches to study inclusion: using a single reviewer, dual review with internal resolution, dual review with third party resolution, or crowdsourcing with three or more reviewers. If you have used just dual review with internal resolution but there was no need for 3rd party resolution, make a mention. In case you did not mention in limitations:

- Introducing some bias,
- Requires highly trained expert in topic to start the search
- Time-consuming while agreeing on resolution

Response: Thank you for this detailed clarification. We had dual reviewers and an internal reviewer. We have now mentioned this in methods. We do not think this is necessarily a study limitation as majority of systematic reviews have an internal reviewer and not an external reviewer, although the latter is preferable but difficult for study conduct and introduces considerable delays as the reviewer mentions.

STRENGTHS AND LIMITATIONS OF THE STUDY

Page 4, Lines 110-113

Even though it is most probably clear to the authors what is considered to be the strength and what the opposite, to the reader some of the bullet points remain ambiguous, Either split in strength and limitations bullet points, or state the strength or limitation side in each statement as you do later on at Page 10, line 287

Response: Thank you. There are only 5 bullet points. We have now amended them to have strengths and limitations.

Comment: Data sources and search strategy

Page 6, Lines 110-113

You did not define or predefine the MeSH terms you used to search by PICO criterion or you do not mention this, In both cases please make a statement about this and add it here.

For more info on PICO look here

<https://mcw.libguides.com/EBM/PICO#:~:text=PICO%20stands%20for%20patient%2Fpopulation%2C%20intervention%2C%20comparison%20and%20outcomes.&text=Who%20is%20your%20patient%3F&text=What%20do%20you%20plan%20on%20doing%20for%20the%20patient%3F&text=What%20alternative%20are%20you%20considering%3F>

Response: Thank you. We have stated the MeSH terms and the word variants in the methods (1st paragraph) and we have also stated the detailed search criteria in the Supplementary Table S1.

Comment: Page 10, Lines 272-273

...prospective cohort screening study

Please rephrase to our (our own is a pleonasm) cohort study comparing the outcome of the screening of incidence of...Stated as it is, introduces confusion about the study design.

What is a screening study? More precisely, a screening trial is designed to assess or test the efficacy of devices, techniques, procedures, or tests to detect the presence of disease or condition (or risk factors for a disease or condition), usually before there are any symptoms.

What is a cohort study? (KOH-hort STUH-dee) A research study that compares a particular outcome (such as lung cancer) in groups of individuals who are alike in many ways but differ by a certain characteristic (for example, female nurses who smoke compared with those who do not smoke). As your study probably compares the outcome of the screening – please rephrase

Response: Thank you. We have now amended this statement in the manuscript.

Page 10, Line 286

Add a paragraph about future research directions

Response: Thank you. A new statement about future directions now added in this section.